# New Insight into the Mechanism of Drug Release from Poly(d,l-lactide) Film by Electron Paramagnetic Resonance

**DOI:** 10.3390/polym12123046

**Published:** 2020-12-18

**Authors:** Natalia A. Chumakova, Elena N. Golubeva, Sergei V. Kuzin, Tatiana A. Ivanova, Igor A. Grigoriev, Sergey V. Kostjuk, Mikhail Ya. Melnikov

**Affiliations:** 1Department of Chemistry, M. V. Lomonosov Moscow State University, Leninskiye Gory, 1/3, 119991 Moscow, Russia; legol@mail.ru (E.N.G.); ser.12.08@yandex.ru (S.V.K.); tatianaivanovamsu@gmail.com (T.A.I.); melnikov46@mail.ru (M.Y.M.); 2Vorozhtsov Novosibirsk Institute of Organic Chemistry Siberian Branch, Russian Academy of Sciences, Lavrentiev Ave., 9, 630090 Novosibirsk, Russia; grig@nioch.nsc.ru; 3Department of Chemistry, Belarusian State University, Leningradskaya Str. 14, 220006 Minsk, Belarus; kostjuks@bsu.by; 4Research Institute for Physical Chemical Problems, Belarusian State University, 14 Leningradskaya Str., 220006 Minsk, Belarus; 5Institute for Regenerative Medicine, Sechenov First Moscow State Medical University, 119991 Moscow, Russia

**Keywords:** poly(d,l-lactide) films, paramagnetic dopants, electron paramagnetic resonance, mechanism of drug release, mathematical model of the release kinetics

## Abstract

A novel approach based on convolution of the electron paramagnetic resonance (EPR) spectra was used for quantitative study of the release kinetics of paramagnetic dopants from poly(d,l-lactide) films. A non-monotonic dependence of the release rate on time was reliably recorded. The release regularities were compared with the dynamics of polymer structure changes determined by EPR, SEM, and optic microscopy. The data obtained allow for the conclusion that the main factor governing dopant release is the formation of pores connected with the surface. In contrast, the contribution of the dopant diffusion through the polymer matrix is negligible. The dopant release can be divided into two phases: release through surface pores, which are partially closed with time, and release through pores initially formed inside the polymer matrix due to autocatalytic hydrolysis of the polymer and gradually connected to the surface of the sample. For some time, these processes co-occur. The mathematical model of the release kinetics based on pore formation is presented, describing the kinetics of release of various dopants from the polymer films of different thicknesses.

## 1. Introduction

Biodegradable polymers are the class of materials widely used in medicine that degrade in a body environment without producing toxic compounds [1]. An application area of a particular polymer is determined mainly by the rate of its dissolution and/or degradation in vivo. Polymer materials that decompose within a few hours (e.g., hydroxyethylcellulose) are used for the short-term delivery of medical substances [2]. Polymer forms that are stable in the body for several weeks or months (for example, polylactide and poly(lactide-co-glycolide) copolymers) are applied for the production of temporary prostheses [3,4] as well as for the creation of long-acting drug delivery systems [5].

Biodegradable polymers for medical purposes can be loaded with various bioactives—growth factors, anti-inflammatory drugs, etc. [6,7,8,9,10,11]. In this paper, such additives are denoted as dopants. One of the most promising approaches for creating doped polymer materials is supercritical fluid technologies. The property of polymers to swell or dissolve in supercritical solvents is used to encapsulate various bioactives. In a ternary system, “supercritical fluid—polymer—dopant”, a transition of the dopant molecules from the supercritical fluid solution into the swollen polymer occurs. Upon depressurization, the solvent gaseous at ambient conditions leaves the polymer matrix ultimately. Such a technique allows for obtaining the doped polymer free of toxic solvent traces. At present, supercritical CO_2_ (scCO_2_) is the most common supercritical solvent combining mild critical parameters (P_c_ = 74 bar and T_c_ = 307 K) with low toxicity. The techniques based on scCO_2_ are used to encapsulate organic molecules: drugs [12], dyes [13], organometallic compounds (followed by thermal, chemical, or radiation treatment of the material to obtain metal-polymer nanocomposites [14,15]), monomers, and initiators for the production of polymer composites [16,17].

One of the essential characteristics of doped polymers for biomedical purposes is the drug release kinetics in vitro and in vivo [18,19,20,21,22]. Kinetic regularities depend on the nature of the polymer, the dopant, and the dosage form or prostheses’ shape and structure. In the case of medical materials doped with medicinal substances, the goal is achieving linear dependence of a drug release on time, that is, the creation of materials characterized by uniform release. Release from a porous matrix is, to a great extent, determined by diffusion of the dopant molecules through the pores and depends on the pores’ size, shape, and permeability. The factors influencing the release kinetics from micronized polymers are the shape, size, and ability of the particles for self-adhesion [22].

Mathematical description of the release of different substances from matrices of aliphatic polyesters is a complicated task due to many physicochemical factors to consider. The most important are modifications of the material structure during swelling and hydrolysis, changes in the diffusion coefficients of the dopant and hydrolysis products during plasticization of the polymer, complex kinetic patterns of hydrolytic degradation of the matrix, etc. Therefore, some authors use a phenomenological description of the release [23,24]. Therefore, for example, on the basis of experimental data analysis on the release of dopants from samples of various architectonics, the authors of [23] proposed the power law. To describe the kinetic curves of dopant release including the induction period, an additional parameter (the lag time) was included corresponding to the time of onset of the release of the substance [24,25,26]. Another modification of the model considers a sharp initial release (so-called “burst”) [27]. This phenomenon is usually attributed to the washout of the substance from the surface of the matrix, but this is contrary to the fact that the release can last for several days [28]. More modern models take into account changes in the diffusion coefficients of the dopants due to plasticization of the polymer matrix as a result of swelling in water or due to a decrease in molecular weight during hydrolysis (see, for example, [26,29]). Models that take into account diffusion in pores formed in polymer matrices seem to be more correct [30,31]. However, the available physicochemical models are primarily based not only on experimental data but also on speculative assumptions not supported by experiments. In this regard, it remains relevant to obtain information on changes in the polymer matrix structure at the macro and micro levels.

Electron paramagnetic resonance (EPR) spectroscopy is a promising method for the study of the release mechanism. This method allows for observing not only the paramagnetic dopant molecules released into the external environment but also the molecules remaining inside the polymer matrix [32,33,34]. The use of stable nitroxide radicals as paramagnetic dopants allows for observing the mobility of dopant molecules inside the polymer at various stages of the release process. It is known that nitroxide radicals are stable in different polymeric matrices during several months [35,36,37]. In our previous work [35], we demonstrated the possibility of doping polylactide with nitroxide radicals using supercritical carbon dioxide.

A serious obstacle to using EPR spectroscopy to study dopant release from polyester materials is the autocatalytic hydrolysis of these polymers producing aliphatic acids [38,39]. To prevent acidification, it is necessary to use a buffer solution and to replace it regularly, which results in a low concentration of the paramagnetic compound in the external environment. Moreover, due to the high dielectric constant of water, the EPR probe volume should be small. When the number of spins in the probe is less than approximately 10^13^, the EPR spectrum has a low signal-to-noise ratio, which leads to low accuracy in determining the radical’s concentration by the standard method based on double integration of the spectrum. For this reason, most works devoted to the release of spin probes from polylactide or poly(lactide-co-glycolide) matrices contain only qualitative analysis of the dynamics of changes in the EPR spectra.

In this paper, the novel approach for obtaining quantitative information from highly noisy EPR spectra was applied to study the features of the release of paramagnetic dopants—nitroxide spin probes, including spin-labeled diclofenac, from poly(d,l-lactide) films to aqueous media. In [35], we show the effectiveness of this method for investigation of the release regularities of spin probes from micronized poly(d,l-lactide). Unfortunately, modeling of the release process from micronized polymer forms is rather complicated. The best way to determine the fundamental regularities of the release kinetics determined by the nature of the polymer and the drug is to investigate the structures with easily reproduced geometry—cylindrical forms, tablets, or films. The last are the most promising since, in this case, the diffusion problem can be regarded as a one-dimensional. It is the release from polymer films that was studied in this work.

The accurately measured release curves were compared with the changes in the rotational mobility of paramagnetic molecules inside the polymer during polymer swelling and hydrolysis. The use of pH-sensitive spin probes allowed for revealing local acidity inside the polymer. Based on the results obtained, one can conclude that the foreground of the dopant release from poly(d,l-lactide) is diffusion through the liquid filling the pores inside the polymer; the contribution of dopant diffusion through the polymer matrix is negligible. The mathematical model of the release kinetics, including two stages of pore formation, was developed.

## 2. Materials and Methods

### 2.1. Materials and Substances

End-caped poly(d,l-lactide) (PDL 04 and PDL 02) was purchased from PURASORB (Netherlands). Stable nitroxide radicals 4-hydroxy-2,2,6,6-tetramethylpiperidine-1-oxide (TEMPOL) and 4-oxo-2,2,6,6-tetramethylpiperidine-1-oxide (TEMPONE) from Sigma-Aldrich were used without further purification. The pH-sensitive spin probes 4-amino-2,2,5,5-tetramethyl-2,5-dihydroimidazole-1-oxyl (ATI) and 5,5-dimetyl-4-dimetylamino-2-ethyl-2-(4-pyridyl)-2,5-dyhydroimidazole-1-oxyl (DPI) (purity > 96%) were synthesized according to experimental procedures described in [40,41], respectively. Spin-labeled bioactive diclofenac (sl-DCF) (purity > 95%) was provided to us by Dr. T. Kalai (University of Pécs, Pécs, Hungary); the methodic of synthesis is given in [35]. The structures of the paramagnetic molecules are presented in Figure 1.

Toluene (Komponent Reaktiv, Moscow, Russia) was distilled over metallic sodium. Phosphate-buffered saline (PBS) was prepared by dissolving the tablets from Pushchinskiye Laboratorii, Moscow region, Russia, in distilled water. pH of the prepared PBS was monitored using a pre-calibrated pH meter/millivoltmeter. Chemically pure carbon dioxide (99.998% grade, NIIKM Ltd., Moscow, Russia) was applied without additional purification.

### 2.2. Samples Preparation

In the first step, the SCF mini-laboratory [42] was used to fabricate porous polymeric matrices of PDL 04 and PDL 02 impregnated by paramagnetic spin probes in scCO_2_ solution. A detailed procedure for the preparation of polymeric porous materials is described in [2]. The supercritical fluid process’s optimal parameters are as follows: temperature 41–43 °C, pressure 160 MPa, holding time 8 h, and pressure release time 50–70 min. Earlier, we showed [43] that the used method allows for forming the samples with uniform distribution of the dopant in the polymer at the macroscopic and molecular levels. The dopant concentration in the polymer was (1–5) × 10^−5^ mol/g in order to reduce the dipole–dipole distortion of EPR spectra. In the case of DPI/PDL 02, the dopant concentration was 3.6 × 10^−6^ mol/g. In the second step, the porous matrices doped with nitroxides were subjected to grinding in a knife mill, sieving through a sieve with a mesh size of <100 μm and subsequent hot (333 K) pressing under a pressure of 2 t/sm^2^ for approximately 30 s (Specac Atlas Manual Hydraulic Press equipment (25 t)). The particles had broad size distribution; however, the predominant fraction was particles with 40–60 μm (48%). To create a film with a thickness of 200–240 μm, 0.30–0.35 g of powder was used.

Quality control of the polymer films was carried out by optical and scanning electron microscopy. Optical microscopy experiments were performed using the Soptop CX40M metallographic microscope, NINGBO SUNNY INSTRUMENT CO., LTD., Zhejiang, China. The dry and swollen films’ surfaces were visualized by the JCM—6000 Neoscope scanning electron microscope, JEOL Ltd., Tokyo, Japan, without gold sputtering. The optimal accelerating voltage not deforming the samples was found to be 10 kV. It is necessary to note that a large number of samples were analyzed and that the most typical images are presented in the article.

### 2.3. Release Experiments

Film fragments of sizes 25–40 mm^2^ were placed into 0.5 mL of PBS (pH 7.3) and kept in orbital shaker-incubator ES-20 (Biosan, Latvia, 250 rpm) at 310 K. Regular (once a day or two) change of the solution was performed to avoid acidification due to hydrolytic degradation of poly(d,l-lactide) [38,39]. The dopant release was controlled by determining of the number of radicals in the samples (approximately 5 μL) taken from the external medium at predetermined times (see below). Due to the replacement of PBS, the release kinetics was measured in differential form. The cumulative kinetic curve was be calculated by summation of the released radical amount.

For studying processes inside the films during swelling and hydrolysis of the polymer, the samples were periodically removed from the liquid and dried with a napkin. Then, the content of paramagnetic substance was measured, and the shape of the EPR spectrum of radicals in the films was analyzed. The structure of the surface and the near-surface layer of the swollen film were studied by SEM and optical microscopy, respectively.

### 2.4. EPR Spectroscopy

#### 2.4.1. Spectra Recording

EPR spectra were recorded by EPR spectrometer Bruker EMX-500 (Bruker, Karlsruhe, Germany). The thermostatic device of Bruker was used. Liquid samples were placed into glass tubes with an inner diameter of 0.9 mm or 1.6 mm; the samples’ height did not exceed 4 mm. Film fragments were studied in quartz tubes with an inner diameter of 4.0 mm. Spectra were recorded at microwave radiation power, not leading to signal saturation.

#### 2.4.2. Numerical Analysis of EPR Spectra

The amount of paramagnetic substance in the polymer samples was defined using the standard method of EPR spectra double integration [44]. The number of radicals in liquid probes cannot be determined using this method for two reasons. Firstly, the outer solution replacement leads to a low concentration of radicals. Secondly, a high dielectric constant of water does not permit studying samples exceeding 4–5 μL. As a result, the average number of radicals in the liquid probe did not exceed 10^12^–10^13^; the EPR spectra of such samples have a meager signal-to-noise ratio (see Figure 2). That is why our team’s method developed [45] was applied for the spectra’s quantitative analysis. The approach is based on convolution of the experimental EPR spectra with the spectrum of the same system “radical—solvent” characterized by a high signal-to-noise ratio. As an example, Figure 2 demonstrates the result of convolution of the EPR spectrum of a liquid probe containing TEMPOL with the spectrum of this radical in PBS. It is seen that a much higher signal-to-noise ratio characterizes the convolution function, and that its intensity can be determined with a smaller error than one of the noisy EPR signal.

EPR spectra of nitroxides in swollen polymer samples are a combination of a broad EPR spectrum of corresponding radicals in dry polymer and a narrow triplet signal. To determine the contribution of the latter signal, the following procedures were used. If the narrow spectrum’s intensity exceeded that of the broad spectrum, subtraction of the spectrum of the radicals in a dry polymer from the spectrum of the swollen sample was performed. Fourier transform of the swollen sample spectrum was used when the narrow spectrum intensity was low. The Fourier image’s absolute value reveals a distinct periodic pattern arising from the narrow triplet signal (see Figure 3). This pattern can be described as follows [46]:(1)$$ℱ\left[I_{EPR}\right]\left(\omega \right)=C\cdot \omega \cdot \exp\left(-b_{G}\omega^{2}\right)\cdot \exp\left(-b_{L}\omega \right)\cdot \frac{\left|1+2\cos\left(a\omega \right)\right|}{3}$$
where $I_{EPR}$ is the *EPR* spectrum, a is the hfi (hyper fine interaction) constant, $b_{G}$ and $b_{L}$ are constants depending on the spectrum shape, and C is a relative integral contribution of the narrow triplet signal in the *EPR* spectrum.

Fitting a periodic pattern of the Fourier image in the region corresponding to the narrow triplet signal allows for calculation of the contribution of this signal to the EPR spectrum of the swollen polymer film.

## 3. Results and Discussion

### 3.1. Physicochemical Characterization of the Samples

The number-average molecular weight (M_n_), weight-average molecular weight (M_w_), and polydispersity (Đ) were determined by size exclusion chromatography using an Ultimate 3000 Thermo Scientific apparatus with Agilent PLgel 5 μm MIXED-C (300 × 7.5 mm) and one precolumn (PL gel 5 μm guard 50 × 7.5 mm) thermostated at 30 °C. The detection was carried out using a differential refractometer as well as diode array detector. Tetrahydrofuran was eluted at a flow rate of 1.0 mL/min. The calculations of molecular weight and polydispersity were carried out using polystyrene standards (Agilent Technologies, Santa Clara, CA, USA). The obtained values of M_n_ were then corrected by a factor of 0.58 [47]. The molecular weight characteristics of the polymers are presented in Table 1. It is seen that the proceeding of the polymers in scCO_2_ did not lead to significant changes in these parameters.

In Figure 4, the SEM images of dry polymeric films are presented. It can be seen that the film is homogeneous at the micro level.

### 3.2. Kinetic Regularities of the Dopants Release from the Poly(d,l-lactide) Films

PR spectra of nitroxide radicals are triplets due to the hyperfine interaction of the unpaired electron with ^14^N nucleus (natural abundance 99.6%, spin equals 1). Due to the high spatial anisotropy of hfc constant, the spectrum shape is susceptible to the rotational mobility of paramagnetic molecules [48]. Figure 5 shows the EPR spectra of TEMPOL in PBS at 298 K and in PDL 02 at 298 K and 100 K. The spectra of all used radicals in PBS and within PDL 04 and PDL 02 are similar to the presented ones. It is seen that the spectra in poly(d,l-lactide) significantly differ from the spectrum in PBS. Because of the low mobility of radicals in the polymer matrix, the anisotropy of g- and hfc-tensors manifests in these spectra. It is necessary to note that the spectrum of the radicals in the polymer recorded at room temperature is close to that recorded at 100 K in the absence of rotational mobility of paramagnetic molecules (in the rigid limit). This fact indicates that the mobility of small compact TEMPOL molecules in PDL 02 is low (the rotational correlation time is about 10^−7^–10^−8^ s). The low mobility of dopant molecules shows that the free volume of PDLLA in glass state (T_g_ is approximately 323–326 K [49]) does not exceed the size of TEMPOL molecules (r is approximately 3.4 Å [50]). The spectrum of the radicals in PBS consists of three narrow lines because of averaging the magnetic anisotropy resulting from the rapid rotation of the paramagnetic molecules.

One of the severe problems which can arise in the case of supercritical impregnation of a polymer is the nonuniform distribution of dopant molecules in the polymeric matrix. To determine local concentration of paramagnetic molecules, the approach proposed in [48] was used. The method is based on analysis of the broadening of the rigid limit spectrum due to the dipole–dipole interaction of paramagnetic molecules. It was found that the EPR spectra of all used paramagnetic substances in all formed samples are not broadened, so there is no reason to assume any significant irregularity of the radical’s distribution in the polymer.

Figure 6 shows the kinetic curves corresponding to the release of different paramagnetic substances from PDL 02 and PDL 04 films into PBS. It is seen that the release of all studied dopants from PDL 02 started immediately after immersion of the samples into the liquid. In contrast, all dopants, regardless of their size and structure, stayed inside the PDL 04 film during ca. 90 days. Simultaneously, destruction of the PDL 04 samples was visually observed, so release of the dopant is most likely caused by erosion of the polymer. In several cases, a small amount (4–7%) of radicals was released from PDL 04 during the first 5–7 days after sample immersion into PBS. Such an initial release (burst) was mentioned in many publications; the cause of this phenomenon is still being discussed [28,51,52].

Several reasons can cause the observed essential difference in the kinetic regularities of the dopant release from PDL 02 and PDL 04 films; the most probable ones are the difference in the polymers’ swelling rate and the pore formation processes. The rate of polymer swelling is characterized by the swelling index (SI), which is the ratio of water uptake to the mass of the dry sample. EPR spectroscopy provides an additional possibility of studying changes in the polymer structure during swelling and hydrolysis. Figure 7 shows the EPR spectra of TEMPOL in PDL 02 film recorded at different periods of swelling. It is seen that the spectra of the swollen samples are a combination of a broad EPR spectrum of radicals in dry polymer and a narrow triplet signal for which intensity increases with time. This signal is a spectrum of radicals localized in regions of the polymer into which water has penetrated. During the first 60 min, the shape of the narrow triplet changes (Figure 7a). It seems that, in the initial period, a gradual increase in the probe mobility was caused by diffusion of water into the surface layer of the sample and gradual increase in the free volume of the polymer in swelling regions. Subsequently, the shape of the narrow triplet becomes close to that of the TEMPOL spectrum in PBS, so the main contribution to the triplet signal is made by paramagnetic molecules localized in pores filled with liquid. The numerical analysis reveals that, at all stages of swelling (except for the first 60 min), the spectra can be presented as a sum of the radicals’ spectrum in dry polymer and the narrow triplet signal.

Determination of the local pH inside the swollen polymer was performed using pH-sensitive nitroxide radicals ATI and DPI. It is known that the distances between the components of the EPR spectra depend on the part of protonated paramagnetic molecules [40]. The calibration curve for ATI is published in [25], and that of DPI is given in Figure 8.

It was found that local pH in the pores of PDL 02 and PDL 04 films was less than 2.5 from the start of the observation of the probe signal in the pores. Hence, we can conclude that most of the pores were formed due to autocatalytic hydrolysis of PDLLA. It is known that this process is accompanied by local acidification of the polymer matrix [32,53]. Figure 7 shows that ca. 60% of TEMPONE molecules release from PDL 04 film into PBS. The remaining ca. 40% of radicals convert into nonparamagnetic species due to disproportionation in acidified medium inside the swollen polymer’s pores [54].

Figure 9 shows the time dependences of the polymers’ swelling indexes and of the part of radicals localized in the pores. It is seen that both PDL 02 and PDL 04 swell in PBS, and the formation of the pores accompanies the swelling. The developed pore system was also visualized by optical microscopy (Figure 10). It is necessary to note that both PDL 02 and PDL 04 films become opaque after several hours in PBS. This fact confirms pore formation with a size close to the visible light wavelengths (400–780 nm).

The swelling index and the part of the radicals localized in the pores of PDL 04 increase simultaneously during observation (Figure 9). These values for PDL 02 increase simultaneously during 4–5 days; after that, the swelling index increases, but the high mobile radicals’ fraction remains constant. The dopant transition rate from the PDL 02 matrix to the pores after five days becomes equal to the dopant removal rate from the pores to the outer solution. Based on the above observations, one can assume that the pores in PDL 02 connect with the surface (open pores), while pores in PDL 04 do not (closed pores). It is in agreement with stagnation of the dopant release from PDL 04 for 90 days. SEM data confirm this assumption. The numerous pores connected with the surface arise in 2 h after immersion of the PDL 02 films into PBS (Figure 11). In contrast, the surface of PDL 04 films kept in PBS for 90 days remained smooth despite clearly recognizing the samples’ swelling. On the 101st day, the areas containing numerous open pores can be seen on the PDL 04 film’s surface. At that time, rapid release of the dopant was observed.

Considering the experimental facts, one can conclude that the transport of dopant molecules occurs basically through pores filled with liquid and connected with the sample’s surface. To prove this hypothesis, the following experiments were performed. The fragments of PDL 04 films containing TEMPONE or sl-DCF were placed into PBS and in parallel to the solution of ascorbic acid (AA) in PBS. It is known that AA is one of the main reducing agents in living bodies which transforms nitroxide radicals to hydroxylamines [55]. AA is often used for investigation of nitroxide stability in different matrices [56,57]. Figure 12 shows that the radical’s decomposition rate inside the swollen polymer does not depend on AA’s presence in the outer liquid. Hence, the AA molecules do not penetrate inside the sample during swelling of the polymer. The AA concentration was chosen to be 5 × 10^−4^ g/mL; 1% of it was sufficient to kill all paramagnetic molecules in the polymer. In contrast, the immersion of partially swollen PDL 02 film into the AA solution resulted in rapid decrease of triplet signal intensity. Obviously, in this case, AA diffused into the open pores and reacted with the nitroxides.

### 3.3. Mathematical Modeling of the Release Process

The development of a mathematical model describing the kinetics regularities of the dopant release from PDLLA into aqueous media is one of the most critical challenges in understanding the mechanism of biodegradation. A reliable model would allow the creation of materials with the required characteristics. The development of the mathematical model turned out to be quite tricky because of many processes that coincide during swelling and degradation of the polymer, such as diffusion of water, formation and evolution of pores, hydrolysis of ester bonds, and diffusion of a dopant through the polymer matrix and in its pores, etc. To date, most of the experimental results concerning the change of polylactide structure during swelling and hydrolysis were obtained using chromatography analysis that is the molecular weight change control [29,58,59]. For this reason, most of the mathematical models describing the dopant release from polylactide to the aqueous medium are based on the diffusion of dopant molecules through swollen and partially hydrolyzed polymer matrices [26,60,61]. Our experiments unambiguously show that diffusion of the dopant molecules is mainly caused by dopant release through the pores filled with liquid not through the polymer matrix. Hence, the mathematical model of the release regularities should be based on formation of the pores.

In the present paper, we announce the mathematical model of a dopant release from PDL 02 films. In this case, the release is controlled by swelling and hydrolysis of the polymer unlike PDL 04, where the release is controlled by erosion of the polymer matrix. Figure 13 shows the differential kinetic curves of the dopant release from PDL 02 films normalized to the initial content of the paramagnetic substance in the samples. For further analysis, the kinetic curves were approximated by polynomials of 5–6 degrees.

The kinetic curve for the 50 μm sample has no notable features (Figure 13b). The kinetic curves for the 200 μm samples are functions with extrema; they have both minimum and maximum. The minimum observed on 5–7 days indicates several processes; one of them leads to a slowdown, and the other one leads to an acceleration of the release rate. The maximum observed at 17–18 days is apparently due to the accelerated release of the dopant and simultaneous decrease in the total amount of the paramagnetic molecules inside the polymer due to their irreversible removal from the system.

Thorough analysis of the SEM images of the PDL 02 films surface allows for establishment of a reason for the minimum on the kinetic curves. Table 2 shows the average number of pores on the surface corresponding to different sample holding times in liquid. It is seen that, after six days, the concentration of surface pores decreased. After that, on days 9 and 11, many pores connected with the surface and characterized by a complex internal structure appeared. The pore closing (pore healing) on the surface of aliphatic polyesters after keeping them in an aqueous medium is discussed in the literature [51,62,63]. This phenomenon is explained by local plasticization caused by penetration of water into the near-surface area of the polymer. Optical microscopy revealed a significant number of non-surface (closed) pores inside PDL 02 film, which was kept in PBS for six days. A large number of opened (connected with the surface) pores on 9–11 days, apparently, were formed by the connection of the closed pores with the surface of the film. Based on the observations, two stages of formation of the pores can be indicated. Therefore, the dopant release can be divided into two phases: release through surface pores, which are partially closed with time, and release through pores formed inside the polymer matrix due to autocatalytic hydrolysis of the polymer and gradually connected to the surface of the sample. For some time, these processes cooccur.

Mathematical modeling of the first stage of the release is a difficult task because of the simultaneous occurrence of many processes—diffusion of water into the polymer, hydrolysis of polymer chains, diffusion of soluble oligomers to the outside, formation and growth of pores, closure of part of the pores on the surface, etc. Many authors have proposed approaches describing these processes separately [59,64,65,66,67], but the unified mathematical model has not been worked out. We describe the first stage release as Fick diffusion without detailed consideration of each process. To account for changes in the polymer matrix during swelling and hydrolysis, we introduce the diffusion coefficient’s dependence on time and coordinate *D_I_(x*,*t)*:(2)$$D_{I}\left(x,t\right)=D_{0}\exp\left(-\lambda \left(\frac{L}{2}-\left|x\right|\right)\right)\left(1-f\cdot e^{-\mu t}\right),$$
where *L* is the film thickness, *x* is the distance between the point under consideration and the center of the film, and subscript *I* implies the first stage of the release process. Parameters *D**_0_*, *λ*, *f*, *μ* characterize the processes that occur in the early stages of polymer swelling and hydrolysis. Since some of the pores which were formed at the first stage do not close but continue to grow deep into the sample, dependence (2) was taken into account when describing the experiment over the entire time.

Modeling the release kinetics at the second stage was performed using a diffusion model proposed in [31] and based on the formation of pores connected with the surface and filled with liquid. According to this model, the dopant molecules diffuse through the polymer matrix with a diffusion coefficient *D_S_*, penetrate the pores, and diffuse in the liquid inside the pores with a diffusion coefficient *D_L_*. Distribution of the low-molecular substance between the liquid phase located in the pores and the polymer phase is described by the distribution constant.
(3)$$\kappa =\frac{C_{L}\left(x,t\right)}{C_{S}\left(x,t\right)}=const$$

The latter hypothesis was not studied experimentally in [31], but our experiments’ results confirm its reliability. One of the basic parameters of the model [31] is the porosity function φ(t*)* which is the volume fraction of the pores formed at a particular point in time. According to our experimental observations, this function should be interpreted as the volume fraction of the pores connected with the sample’s surface (open pores). In [31], the porosity function was presented as follows:(4a)$$\varphi \left(t\right)=\varphi_{0}+\left(1-\varphi_{0}\right){\left(1-e^{-kt}\right)}^{2},$$
where *φ**_0_* is the initial porosity; thus, the formation of pores begins immediately after immersion of the sample in an aqueous medium. To describe the second stage of the release, the porosity function should be shifted in time. Besides, to take into account the hydrolysis’s autocatalytic character, we changed the rate of the function increase by changing the exponent:(4b)$$\varphi \left(t\right)=\{\begin{array}{c} 0,\;t<t_{lag}\\ {\left(1-e^{-k\left(t-t_{lag}\right)}\right)}^{\frac{3}{2}},\;t>t_{lag} \end{array}.$$
where *t_lag_* means the time when the second stage of release starts.

Penetration of dopant molecules into the pores and dopant release as these pores gradually connects with the surface are described in [68] by the average diffusion coefficient:(5)$$D_{II}\left(t\right)=\frac{\left(1-\varphi \left(t\right)\right)D_{S0}e^{\alpha k_{W}t}+\kappa \varphi \left(t\right)D_{L}}{1-\varphi \left(t\right)+\kappa \varphi \left(t\right)},$$
where *D_L_* is the effective diffusion coefficient of the dopant in the pore which also reflects complex geometry of the pores, *D_S_*_0_ is the diffusion coefficient of the dopant in a dry polymer, *k_W_* is the rate constant of hydrolysis of the polymer chains, *α* is the empirical parameter connecting the diffusion coefficient of the dopant in the polymer matrix with the average molecular weight of the polymer, and subscript *II* means the second stage of the release process.

Under the conditions of our experiments, the contribution of the dopant diffusing through the polymer matrix to the release process is significantly lower (1–2 order of magnitude) than the contribution of the dopant diffusing through the pores. Therefore, the summand $D_{S}\left(t\right)=D_{S0}e^{\alpha k_{W}t}$ can be neglected. The values of φ(t) turn out to be relatively small throughout the experiment; therefore, we can only consider the first term in the Taylor series:(6)$$\varphi \left(t\right)={\left(1-e^{-k\left(t-t_{lag}\right)}\right)}^{\frac{3}{2}}\approx {\left(k\left(t-t_{lag}\right)\right)}^{\frac{3}{2}}=k^{\frac{3}{2}}{\left(t-t_{lag}\right)}^{\frac{3}{2}},\;t>t_{lag}$$
(7)$$D_{II}\left(t\right)=\frac{\kappa D_{L}k^{\frac{3}{2}}{\left(t-t_{lag}\right)}^{\frac{3}{2}}}{1-\varphi \left(t\right)+\kappa \varphi \left(t\right)}\approx \kappa D_{L}k^{\frac{3}{2}}{\left(t-t_{lag}\right)}^{\frac{3}{2}},\;t>t_{lag}.$$

From Expression (7), it is seen that the term $A=\kappa D_{L}k^{3/2}$ should be considered a unified parameter.

When developing the mathematical model, the following additional assumptions were taken into account:(1)The dry polymer film is structurally uniform.(2)The dopant is distributed uniformly in a dry sample.(3)Diffusion through the side surface of the film is negligible.(4)Perfect-sink conditions are used.

Based on the assumptions described above, the mathematical model has the following form.
(8)$$\frac{\partial C\left(x,\;t\right)}{\partial t}=\frac{\partial }{\partial x}\left[D\left(x,\;t\right)\frac{\partial C\left(x,\;t\right)}{\partial x}\right]$$
(9)$$D\left(x,t\right)=D_{0}\exp\left(-\lambda \left(\frac{L}{2}-x\right)\right)\left(1-f\;e^{-\mu t}\right)+\kappa D_{L}\varphi \left(t\right)$$
(10)$$C\left(x,\;0\right)=C_{0}$$
(11)$$C\left(\frac{L}{2},\;t\right)=0$$
(12)$${\frac{\partial C\left(x,\;t\right)}{\partial x}}_{x=0}=0$$
$$x\in \left[0,\frac{L}{2}\right],\;t\in \left[0,\;t_{max}\right]$$

After solving the differential equation, the release rate normalized to the initial amount of dopant in the sample (*N*_0_) can be calculated as follows:(13)$$\frac{1}{N_{0}}\frac{dN_{out}}{dt}=\frac{2S}{N_{0}}\int_{0}^{\frac{L}{2}}\frac{\partial C\left(x,\;t\right)}{\partial t}dx=\frac{2S}{N_{0}}\int_{0}^{\frac{L}{2}}\frac{\partial }{\partial x}\left[D\left(x,\;t\right)\frac{\partial C\left(\chi ,\;t\right)}{\partial x}\right]dx=\frac{2D\left(\frac{L}{2},\;t\right)}{LC_{0}}{\frac{\partial C\left(\chi ,\;t\right)}{\partial x}}_{x=\frac{L}{2}}$$

On the whole, the diffusion model is characterized by the following parameters: *D*_0_, *λ*, *μ*, and *f* are empirical parameters describing the first stage release; *t_lag_* defines the time when connection of the internal pores with the surface starts; and parameter *A = κD_L_k*^3/2^ reflects the intensity of dopant release at the second stage. The last parameter characterizes the affinity of the dopant for the matrix and the hydrolytic stability of the matrix.

The results of modeling the experimental differential kinetic curves are presented in Figure 14. Figure 15 shows the corresponding descriptions of the integral kinetic curves. It is seen that, despite a slight deviation of the theoretical differential curves from the experimental ones, the description of the integral release curves is within the registration errors. The results indicate that the difference in the shape of the kinetic curves for 50 μm and 200 μm films is due to different contributions of the dopant release through the pores formed in the near surface layer (first stage of pore formation) and through the pores formed inside the polymer and then connected with the surface (second stage of pore formation).

The parameters of the modeling are presented in Table 3. For system 1, two independent release experiments were performed. It can be noted that the optimal value of parameter *t_lag_* for system 1 is close to the time when the fraction of mobile radicals reached the limit (Figure 9) and is consistent with the data obtained by scanning electron microscopy (Figure 11). The value of parameter *A* for 200 μm films is approximately 20 times higher than for 50 μm film. Since the value *κ* contributing to this parameter is determined only by the nature of the polymer and dopant, the observed difference reflects greater hydrolytic stability of thin PDLLA films, consistent with published data [68,69,70]. Parameter *D*_0_ reflects the rate of pore formation at the first stage. One can assume from the data of Table 3 a slower surface pore formation in a thin film. The absence of correlation of the parameters *λ*, μ, and *f* both with film thickness and the nature of the radical perhaps indicates differences in the structure of the surface layer caused by manual pressing of the films. Further application of the proposed to various “polymer/dopant” systems would clarify these parameters’ physical meaning.

## 4. Conclusions

EPR spectroscopy combined with modern methods of numerical analysis of spectra allowed for monitoring of the changes in the paramagnetic probe content both in the biodegradable polymeric matrix during hydrolysis and in the external environment simultaneously. As a result, it was found that loss of the probe inside the poly(d,l-lactide) did not mean only its transition to the external environment. In an acidic medium formed during hydrolysis of the polyester, radicals may transform into diamagnetic substances. Registration of the release curves in a differential form allowed for estimation of the pronounced extrema of the release rate on time, making it possible to narrow the range of mathematical models of the release. Joint analysis of EPR spectroscopy, SEM, and water uptake data allowed us to conclude that the central role in the release plays diffusion in pores that arise, grow, or close up in the polymer matrix. Combination and competition of these processes explain the “burst” and lag periods. It was proposed that diffusion through the swollen polymer is negligible. The mathematical model of the release kinetics based on pore formation was performed.

## Figures and Tables

**Figure 1 polymers-12-03046-f001:**
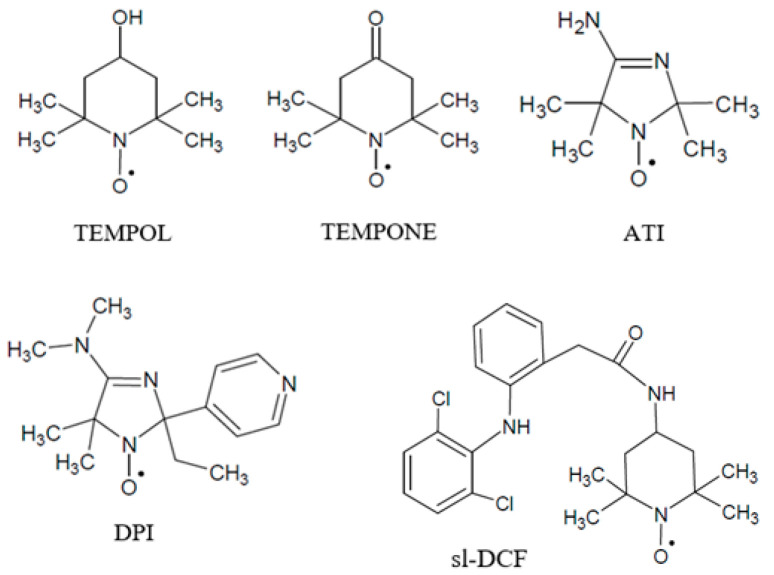
The structures of the paramagnetic compounds.

**Figure 2 polymers-12-03046-f002:**
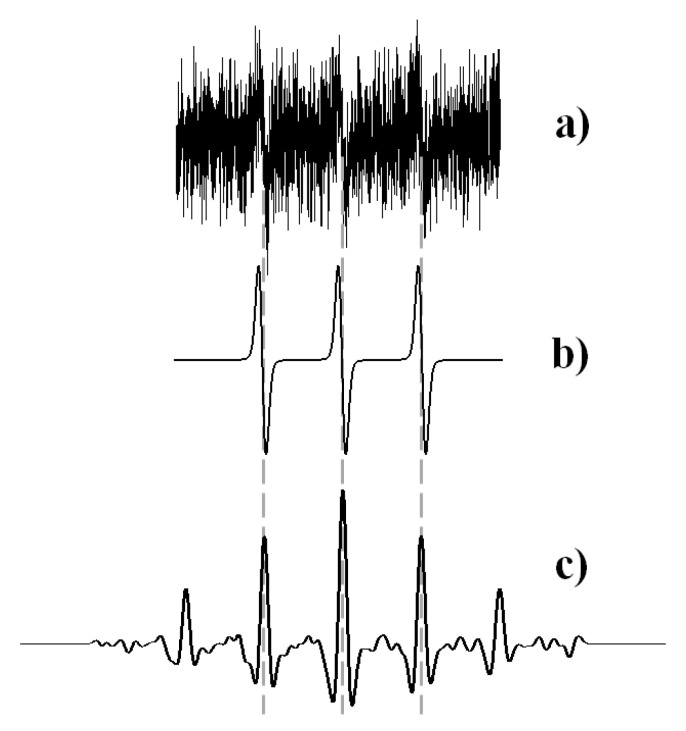
Electron paramagnetic resonance (EPR) spectrum of a liquid probe containing 4-hydroxy-2,2,6,6-tetramethylpiperidine-1-oxide (TEMPOL) (**a**), the spectrum of TEMPOL in PBS (**b**), and the result of their convolution (**c**).

**Figure 3 polymers-12-03046-f003:**
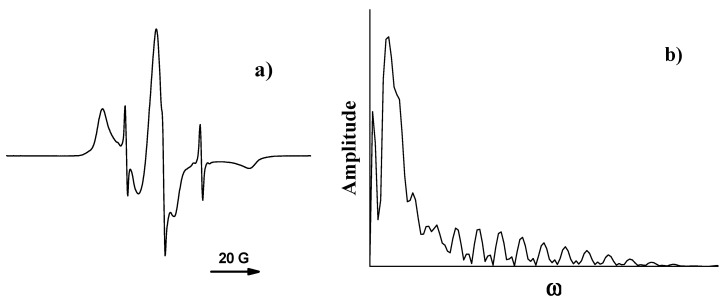
The EPR spectrum of 4-oxo-2,2,6,6-tetramethylpiperidine-1-oxide (TEMPONE) in swollen poly(d,l-lactide) film (**a**) and the absolute value of its Fourier image (**b**): the periodic pattern corresponds to the narrow triplet signal.

**Figure 4 polymers-12-03046-f004:**
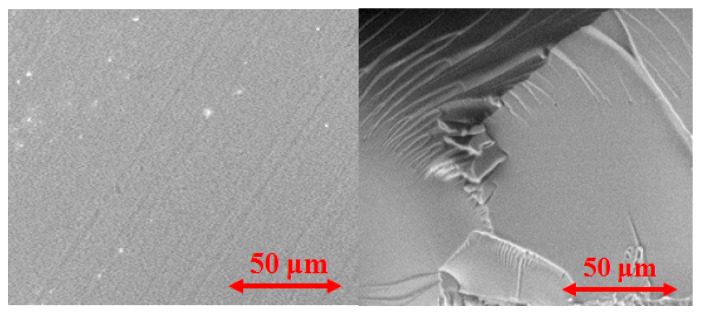
SEM images of dry 200 μm poly(d,l-lactide) (PDL) 02 film: surface (**left**) and chip (**right**).

**Figure 5 polymers-12-03046-f005:**
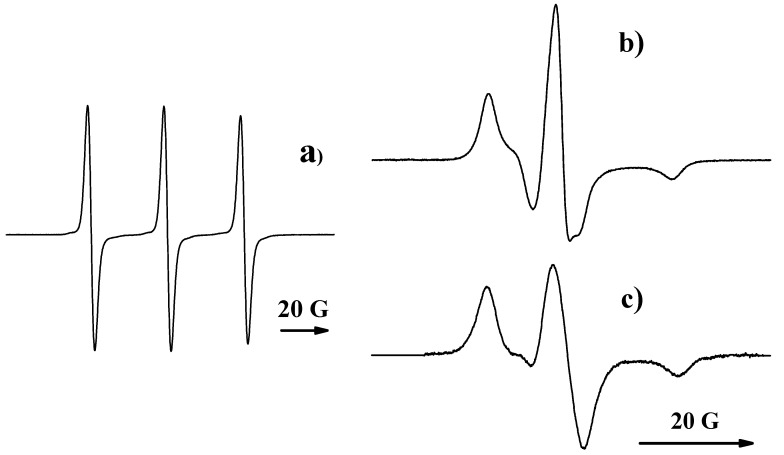
EPR spectra of TEMPOL in PBS at 298 K (**a**) and in PDL 02 at 298 K (**b**) and 100 K (**c**).

**Figure 6 polymers-12-03046-f006:**
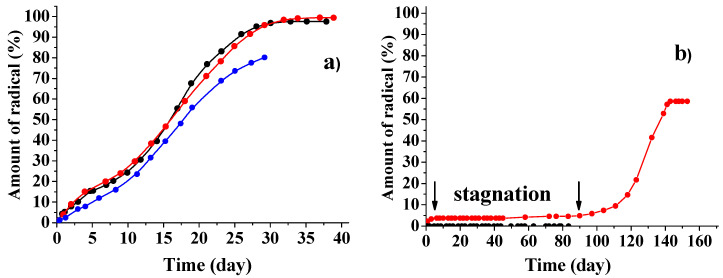
Kinetic curves corresponding to the release of TEMPOL (black and red symbols) and ATI (blue symbols) from PDL 02 films (**a**) and of sl-DCF (black symbols) and TEMPONE (red symbols) from PDL 04 films (**b**) into PBS: measurement errors are 10–12%. The lines are not the result of fitting.

**Figure 7 polymers-12-03046-f007:**
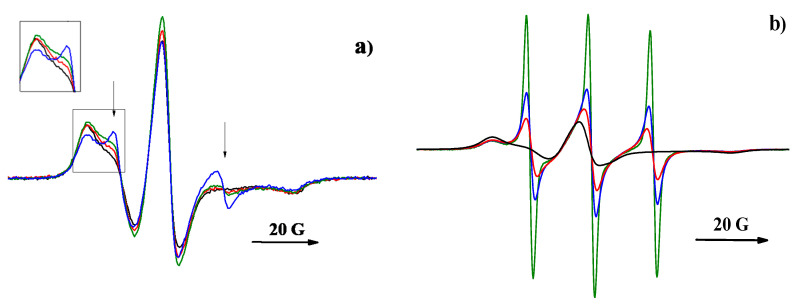
The EPR spectra of TEMPOL in PDL 02 film recorded at various holding times of the sample in solution: (**a**) dry sample (black line), 30 min (red line), 1 h (green line), and 6 h (blue line); (**b**) dry sample (black line), 2 days (red line), 7 days (blue line), and 17 days (green line).

**Figure 8 polymers-12-03046-f008:**
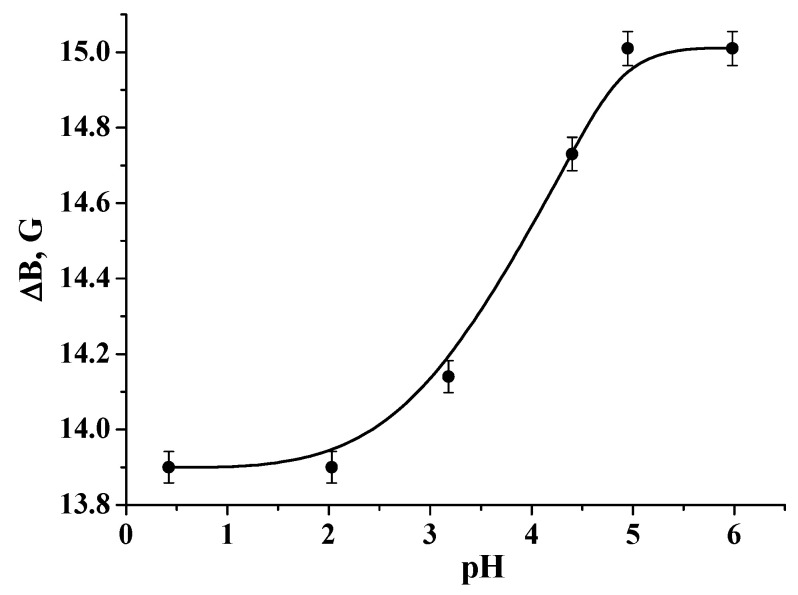
pH dependence of the distance between the central and left components (ΔB) of the EPR spectra of DPI in water solution.

**Figure 9 polymers-12-03046-f009:**
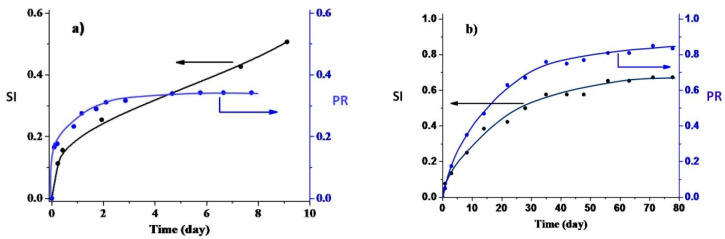
Time dependences of the swelling index (SI, black symbols, measurement errors are 5–7%) and of the part of radicals, localized in pores (PR, blue symbols, measurement errors are 10–12%) for TEMPOL in PDL 02 film (**a**) and for TEMPONE in PDL 04 film (**b**): the lines are not the results of fitting.

**Figure 10 polymers-12-03046-f010:**
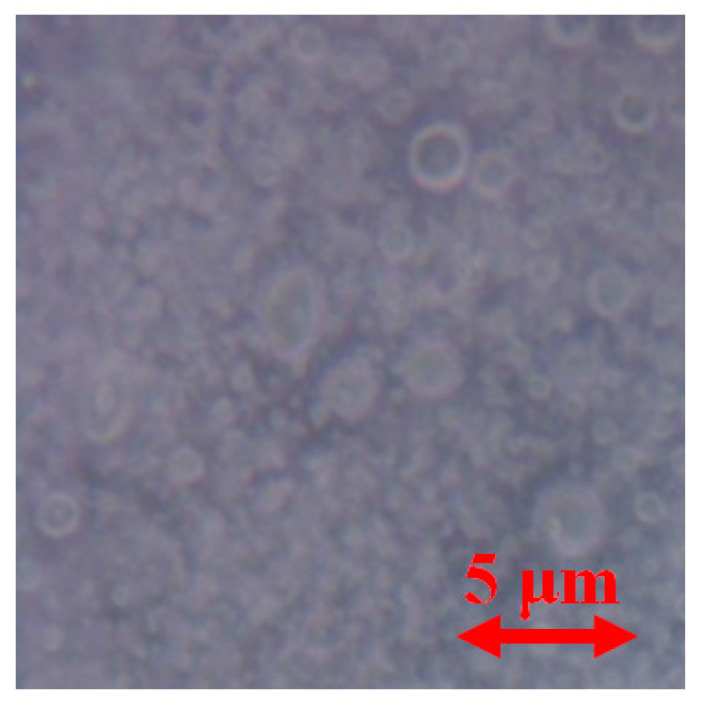
Optical microphotographs of PDL 04 film recorded after 3 weeks of keeping in PBS.

**Figure 11 polymers-12-03046-f011:**
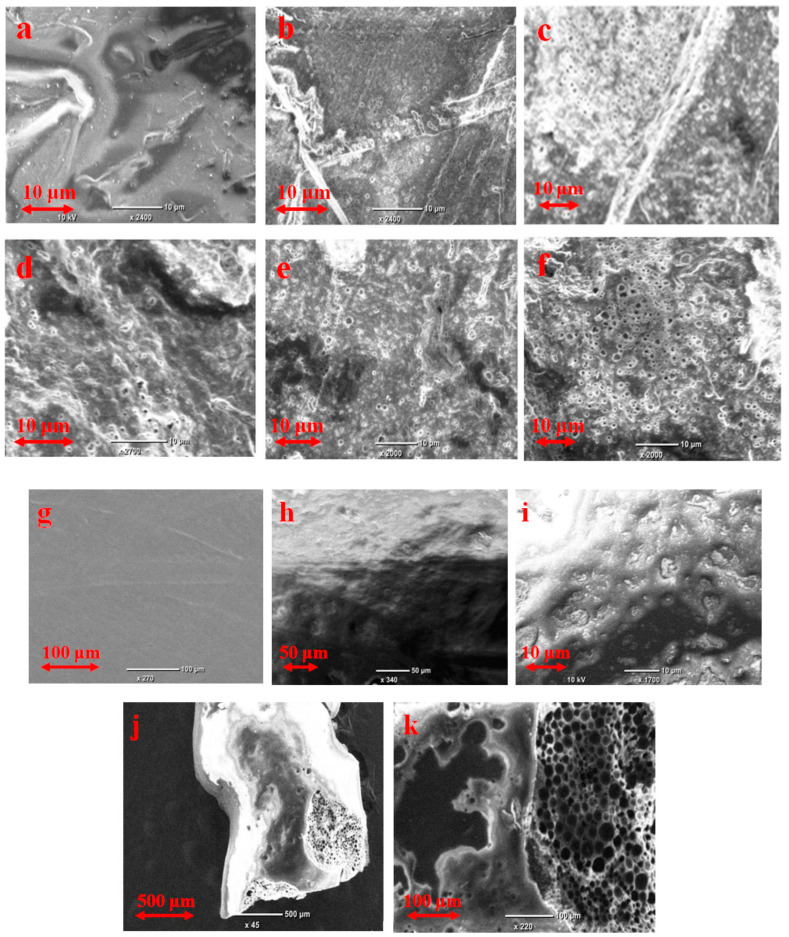
SEM microphotographs of PDL 02 (**a**–**f**) and PDL 04 (**g**–**k**) films recorded after different time of keeping them in PBS: (**a**)—dry film, (**b**)—2 h, (**c**)—4 days, (**d**)—6 days, (**e**)—9 days, (**f**)—11 days, (**g**)—1 day, (**h**)—5 days, (**i**)—90 days, and (**j**,**k**)—101 days.

**Figure 12 polymers-12-03046-f012:**
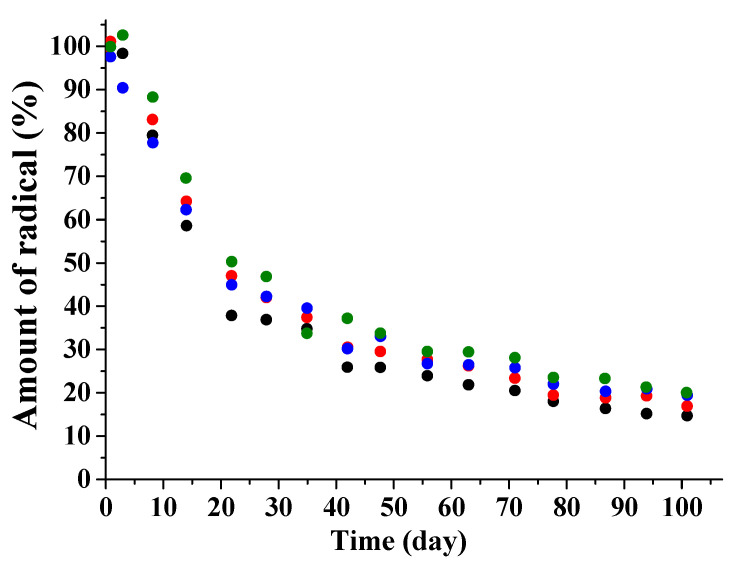
The kinetic curves of the radical’s decomposition inside PDLLA films in the presence and in the absence of AA in the outer liquid: systems “TEMPONE/PBS + AA” (black symbols), “sl-DCF/PBS + AA” (red symbols), “TEMPONE/PBS” (blue symbols), and “sl-DCF/PBS” (green symbols). Measurement errors are 10–12%.

**Figure 13 polymers-12-03046-f013:**
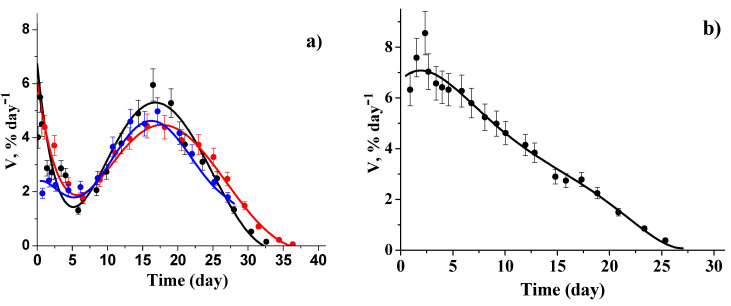
Differential kinetic curves for the dopants release from PDL 02 films: TEMPOL/200 μm films (black and red symbols) and ATI/200 μm film (blue symbols) (**a**), and TEMPOL/50 μm film (**b**). The lines are the result of polynomial fitting.

**Figure 14 polymers-12-03046-f014:**
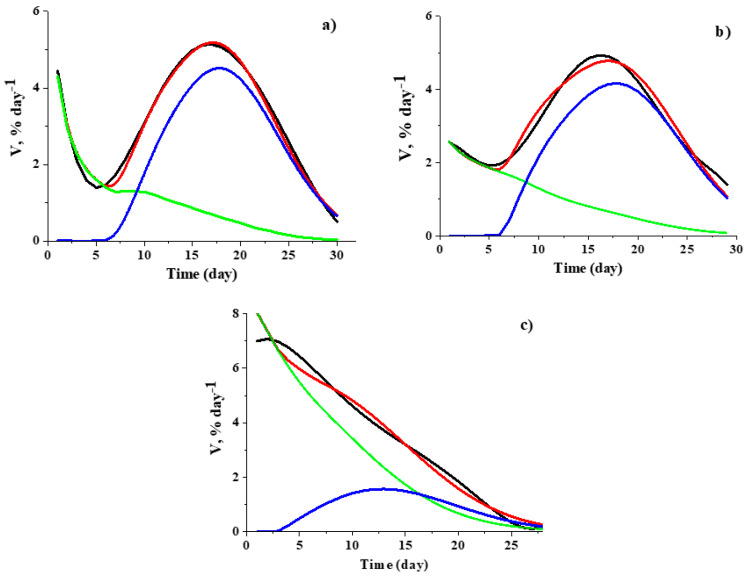
Experimental differential curves (black lines) and corresponding numerical modeling of TEMPOL (**a**) and ATI (**b**) release from 200 μm films and of TEMPOL (**c**) release from 50 μm film: the red lines are the results of modeling, and green and blue lines are contributions of the dopant release through the pores formed at the first and the second stages of pore formation, respectively (see text).

**Figure 15 polymers-12-03046-f015:**
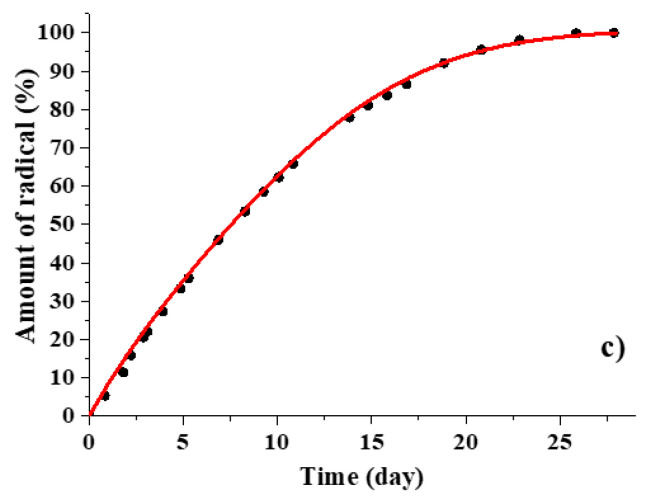
Experimental cumulative curves (dots) and corresponding numerical modeling (lines) of TEMPOL (**a**) and ATI (**b**) release from 200 μm PDLLA films and of TEMPOL (**c**) release from 50 μm film.

**Table 1 polymers-12-03046-t001:** Molecular weight characteristics of the polymers.

	M_n_	M_w_	Đ
PDL 04	14,800	36,100	2.4
PDL 04, after SCF processing	15,200	35,800	2.4
PDL 02	7200	15,600	2.2

**Table 2 polymers-12-03046-t002:** An average number of pores on the surface of the swelling PDL 02.

Time	Average Number of Pores per 100 μm^2^
2 h	16
2 days	7
4 days	8
6 days	4
9 days	8
11 days	10

**Table 3 polymers-12-03046-t003:** The optimal values of the modeling parameters.

System		D_0_, m^2^·Day^−1^	λ, m^−1^	μ, Day^−1^	f	$t_{lag}$, Day	A, m^2^·Day^−5/2^
1	TEMPOL/200 μm	7.5 × 10^−11^	5.0 × 10^4^	1.4	0.5	6.1	1.35 × 10^−11^
1	TEMPOL/200 μm	7.5 × 10^−11^	2.5 × 10^4^	1.0	0.3	5.9	0.94 × 10^−11^
2	TEMPOL/50 μm	2.4 × 10^−11^	0.9 × 10^4^	0.33	1.0	2.6	4.77 × 10^−13^
3	ATI/200 μm	6.9 × 10^−11^	1.4 × 10^4^	0.17	0.9	5.8	1.11 × 10^−11^

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
