# Peer review of "New Insight into the Mechanism of Drug Release from Poly(d,l-lactide) Film by Electron Paramagnetic Resonance"

_polymers, 2020, doi:10.3390/polym12123046_

Round 1
Reviewer 1 Report
- On line 36, the sentence “Biodegradable polymers for medical purposes can contain various dopants – growth factors, anti-inflammatory drugs .. is a bit confusing as the term dopants indicate more of impurities or residual presence while in medical applications, the polymer is loaded with therapeutic moieties or Active Pharmaceutical Ingredients (API). I suggest rephrasing the sentence to become clearer.
- Under the materials and methods, please separate the materials section from the methodologies and procedures. Some of the writing are results of those experiments like SEC result for PLDLA and must be under results and discussed under discussion. The reader will feel lost when reading these parts of the manuscript, and a clear section of materials followed by methods with no results or discussion should be clearly written and presented. For example, on line 134, after describing the methodology, the result is also presented, and figure 2 was also shown, which confuses the reader tremendously.
- Explicit and precise use of terminologies like paramagnetic compounds versus dopants is needed …etc
Author Response
- On line 36, the sentence “Biodegradable polymers for medical purposes can contain various dopants – growth factors, anti-inflammatory drugs .. is a bit confusing as the term dopants indicate more of impurities or residual presence while in medical applications, the polymer is loaded with therapeutic moieties or Active Pharmaceutical Ingredients (API). I suggest rephrasing the sentence to become clearer.
We agree that the mentioned sentence in not quite clear and propose the following rephrasing. “Biodegradable polymers for medical purposes can be loaded with various bioactives – growth factors, anti-inflammatory drugs, etc []. In this paper, such additives are denoted as dopants.”
- Under the materials and methods, please separate the materials section from the methodologies and procedures. Some of the writing are results of those experiments like SEC result for PLDLA and must be under results and discussed under discussion. The reader will feel lost when reading these parts of the manuscript, and a clear section of materials followed by methods with no results or discussion should be clearly written and presented. For example, on line 134, after describing the methodology, the result is also presented, and figure 2 was also shown, which confuses the reader tremendously.
We agree with the reviewer. Both SEC and SEM results were moved to section 3 Results and discussion.
- Explicit and precise use of terminologies like paramagnetic compounds versus dopants is needed …etc
On lines 61-62 we introduce the nitroxide radicals as paramagnetic dopants: “The use of stable nitroxide radicals as paramagnetic dopants…” But for more precise use of terminology we propose the following change in the phrase on lines 73-75. “In this paper, the novel approach for obtaining the quantitative information from highly noisy EPR spectra was applied to study the features of the release of paramagnetic dopants - nitroxide spin probes, including spin-labeled diclofenac, from poly(D,L-lactide) films to aqueous media.”
Reviewer 2 Report
This research is a well written and complex article that combines the EPR spectroscopy with modern methods of numerical analysis of the spectra to evaluate the physical-chemical processess associated with the mechanism of drug release from biodegradable polymeric films. This research was completed with the mathematical modeling of the release processes based on the pore formation.
I have some remarks for the Authors, namely:
- Line 17: Please define firstly the abbreviation EPR: electron paramagnetic resonance, and than use the abbreviation.
- Lines 134-135: Please move the phrase: “As it can be seen in Figure 2, dry polymeric films are homogeneous at the micro-level.” at the Results and Discussion Section.
- The same remark for the Figure 2.
- Minor english revison is required.
Author Response
1. Line 17: Please define firstly the abbreviation EPR: electron paramagnetic resonance, and than use the abbreviation.
We added the full name of the method.
2. Lines 134-135: Please move the phrase: “As it can be seen in Figure 2, dry polymeric films are homogeneous at the micro-level.” at the Results and Discussion Section.
3. The same remark for the Figure 2.
We moved the phrase and the figure to section 3 Results and discussion.
4. Minor english revison is required.
The article was thoroughly edited.
Reviewer 3 Report
The manuscript (polymers-1016410) entitled “New insight to the mechanism of drug release from poly(D,L-lactide) film by electron paramagnetic resonance” is well written and supported with reasonable experiments & outcomes. Indeed, authors are advised to take into the consideration following suggestions.
- Authors are suggested to modify the title which reflects the purpose of this investigation as well.
- Abstract needs to be revised with a clear statement of the aim and objective of the current investigation. It is also advised to briefly include the methodology involve in the investigation and then include the results (with numerical value) obtained from this investigation.
- In the introduction section, it is suggested, little concise theoretical discussion and include contemporary research carried out by other investigators in this area with a highlight of your novelty in this investigation.
- In section 2.1. only discuss the procurement source of materials and exclude the experimental part like size exclusion chromatography detail for molecular weight determination. Include all this experimental discussion as separate section as “Physicochemical characterization of procured sample for quality analysis”
- Authors are suggested to highlight the mechanism of drug release by considering any specific model drug and emphasize the influence of dopant release on the kinetics of drug release as well.
- Authors are advised to highlight the novelty of the current investigation in comparison to your previous work “Paramagnetic bioactives encapsulated in poly(D,L-lactide) microparticules: Spatial distribution and in vitro release kinetics” published recently in the year 2020.
Author Response
1. Authors are suggested to modify the title which reflects the purpose of this investigation as well.
We would not like to modify the title. The present title reflects both novelty of the research and the basic method of the investigation.
2. Abstract needs to be revised with a clear statement of the aim and objective of the current investigation. It is also advised to briefly include the methodology involve in the investigation and then include the results (with numerical value) obtained from this investigation.
We revised Abstract including the main qualitative results of the work. These are the results that are most important for further development of the release processes investigation. As for numerical values, these parameters were obtained for the particular polymer undergoing the particular treatment; they can be used for comparison of different systems only.
3. In the introduction section, it is suggested, little concise theoretical discussion and include contemporary research carried out by other investigators in this area with a highlight of your novelty in this investigation.
We agree with the reviewer. The discussion of the mathematical models of the release kinetic regularities which were worked out by other investigators was added into the Introduction.
4. In section 2.1. only discuss the procurement source of materials and exclude the experimental part like size exclusion chromatography detail for molecular weight determination. Include all this experimental discussion as separate section as “Physicochemical characterization of procured sample for quality analysis”
We agree with the reviewer. Both SEC and SEM results were moved to section 3 Results and discussion.
5. Authors are suggested to highlight the mechanism of drug release by considering any specific model drug and emphasize the influence of dopant release on the kinetics of drug release as well.
The results obtained allow establishing of the basic kinetic regularities of the release processes. It is assumed that release of the specific drug would be similar to release of the similar paramagnetic dopant (molecular size, set of functional groups, ect.) To establish the regularities of the release of the particular drug, one can mark it with a paramagnetic fragment, as it was done in this work with non-steroidal anti-inflammatory drug diclofenac (DCF).
6.Authors are advised to highlight the novelty of the current investigation in comparison to your previous work “Paramagnetic bioactives encapsulated in poly(D,L-lactide) microparticules: Spatial distribution and in vitro release kinetics” published recently in the year 2020.
Our previous work is mainly devoted to the technique of doping polylactide with paramagnetic substances using supercritical carbon dioxide and to possibility of EPR spectroscopy for investigation of the release regularities. The release of dopants from micronized polymer could not be described within the framework of the diffusion model due to the complex geometry of the particles and their partial adhesion in solution. The present work is devoted to study of the polymer films – the simplest object for determining the fundamental regularities of the release kinetics. The mathematical model describing the release kinetics was worked out as a result.
Round 2
Reviewer 1 Report
- English language writing must be heavily checked.
- I am not sure what the authors meant by the NEW insight into the title! The use of EPR in monitoring and predicting drug release from polymers by adding paramagnetic molecules or groups, e.g., nitroxides, to allow detection has already been reported. A large variety of nitroxides (also known as spin probes) with different physicochemical properties is commercially available for that purpose. As such, I am not sure how much this concept is new or novel! Please refer to the references below, which discuss in detail a similar principle as what is presented in this manuscript.
- Mäder K, Bacic G, Domb A, et al. Noninvasive in vivo monitoring of drug release and polymer erosion from biodegradable polymers by EPR spectroscopy and NMR imaging. Journal of Pharmaceutical Sciences. 1997 Jan;86(1):126-134. DOI: 10.1021/js9505105.
- Lurie, D. J., & Maeder, K. (2005). Monitoring drug delivery processes by EPR and related techniques – principles and applications. Advanced Drug Delivery Reviews, 57, 1171-1190. https://doi.org/10.1016/j.addr.2005.01.023
- The statement starting on line 41 must use more recent references. I strongly suggest adding the following more recent references:
- Bayan Alemrayat, Abdelbary Elhissi & Husam M. Younes (2019) Preparation and characterization of letrozole-loaded poly(d,l-lactide) nanoparticles for drug delivery in breast cancer therapy, Pharmaceutical Development and Technology, 24:2, 235-242, DOI: 10.1080/10837450.2018.1455698
- Sun, C., Zou, L., Xu, Y., Wang, Y., Ibuprofen‐Loaded Poly(Lactic Acid) Electrospun Mats: The Morphology, Physicochemical Performance, and In Vitro Drug Release Behavior. Macromol. Mater. Eng. 2020, 2000457. https://doi.org/10.1002/mame.202000457.
- The statement on line 55, starting with "In the case of medical materials doped with medicinal 55 substances, the goal is achieving linear dependence of a drug release on time, that is, the creation of 56 materials characterized by a uniform release" requires a supporting reference. Consider adding the following reference, which discusses these aspects (Drug Development and Industrial Pharmacy, 25 Sep 2018, 44(12):1953-1965 DOI: 10.1080/03639045.2018.1503298)
- On line 71, please use lag time instead of delay time (more common in drug delivery)
- The sentence on Line 108 should start from the beginning of the line.
- Line 136, The dry ….the T is missing
- On line 165, please use the term free radicals instead of radicals.
- The materials and methods section still reports results in it, which are supposed to be under results and discussion. Please rectify.
- Justification and discussion about why the researchers chose the film's geometry and not a particle or cylindrical or tablet form of the polymer. Would the EPR have the ability to measure drugs and radicals movement the same in this case with the same accuracy and sensitivity?
Author Response
- English language writing must be heavily checked.
- I am not sure what the authors meant by the NEW insight into the title! The use of EPR in monitoring and predicting drug release from polymers by adding paramagnetic molecules or groups, e.g., nitroxides, to allow detection has already been reported. A large variety of nitroxides (also known as spin probes) with different physicochemical properties is commercially available for that purpose. As such, I am not sure how much this concept is new or novel! Please refer to the references below, which discuss in detail a similar principle as what is presented in this manuscript.
- Mäder K, Bacic G, Domb A, et al. Noninvasive in vivo monitoring of drug release and polymer erosion from biodegradable polymers by EPR spectroscopy and NMR imaging. Journal of Pharmaceutical Sciences. 1997 Jan;86(1):126-134. DOI: 10.1021/js9505105.
- Lurie, D. J., & Maeder, K. (2005). Monitoring drug delivery processes by EPR and related techniques – principles and applications. Advanced Drug Delivery Reviews, 57, 1171-1190. https://doi.org/10.1016/j.addr.2005.01.023
Of course, spin probe method for studying processes in polymers is a widespread approach, described in many books and reviews. The Prof. Maeder group made a great contribution to the study of processes occurring in polymers in vitro and in vivo, and we cite him in the manuscript (ref. [32-34]. Nevertheless, some questions are opened for discussion. Among them:
- the reason for the decrease in the content of the probe in the polymer matrix;
- the nature of the “burst” period lasting several hours or even days;
- lack of a unified mathematical model for the release of low molecular weight substances from aliphatic polyesters, which allows adequately describing release curves of different types;
-etc.
In the manuscript, the quantitative EPR based on the method of convolution of experimental spectra with a standard spectrum allowed us to monitor the changes in the paramagnetic probe content in the polymer matrix and the external environment simultaneously. This approach was applied for the first time, as we know. Based on these data, we have established that the loss of the probe inside the matrix does not mean only its transition to the external environment. In an acidic medium formed during the hydrolysis of polyesters, radicals and other reactive species may transform to different products, e.g., diamagnetic substances. That is why measuring the paramagnetic probe's content in the polymer matrix does not provide the full information about the release.
Due to the regular exchange of the buffer solution, the release curves were for the first time recorded in a differential form (see Fig.13), that is, as a dependence of the release rate on time. These curves have two pronounced extrema, which makes it possible to narrow the range of mathematical models of the release.
It is well-known that EPR spectra of nitroxides in swollen PDLA (PLGA) samples are a combination of a broad EPR spectrum of corresponding radical in dry polymer and a narrow triplet signal of the mobile radical in a liquid medium. We use the Fourier transform of the swollen sample spectrum to estimate the latter signal's contribution and fount that the fraction of high mobile radicals in the matrix remains constant for a long time. Accordingly, the probe transition rate from the PDL 02 matrix to the pores after five days becomes equal to the dopant removal rate from the pores to the outer solution. This statement, lacking experimental evidence, the authors of [31] used to formulate the mathematical model of release from ZDPF films. Hereinafter we improved this model for simulating our experimental data.
Joint analysis of EPR spectroscopy, SEM, and water uptake data allowed us to conclude that the central role in the release plays diffusion in pores that arise, grow or close up in the polymer matrix. Combination and competition of these processes explain the “burst” and lag periods. Diffusion through the swollen polymer is negligible.
As a result, we proposed a new mathematical model of the release, which adequately describes both the kinetic curves with extrema and monotonic dependences.
- The statement starting on line 41 must use more recent references. I strongly it adding the following more recent references:
- Bayan Alemrayat, Abdelbary Elhissi & Husam M. Younes (2019) Preparation and characterization of letrozole-loaded poly(d,l-lactide) nanoparticles for drug delivery in breast cancer therapy, Pharmaceutical Development and Technology, 24:2, 235-242, DOI: 10.1080/10837450.2018.1455698
- Sun, C., Zou, L., Xu, Y., Wang, Y., Ibuprofen‐Loaded Poly(Lactic Acid) Electrospun Mats: The Morphology, Physicochemical Performance, and In Vitro Drug Release Behavior. Mater. Eng. 2020, 2000457. https://doi.org/10.1002/mame.202000457.
Thank you for the suggestion, the references were added to the manuscript.
- The statement on line 55, starting with "In the case of medical materials doped with medicinal 55 substances, the goal is achieving linear dependence of a drug release on time, that is, the creation of 56 materials characterized by a uniform release" requires a supporting reference. Consider adding the following reference, which discusses these aspects (Drug Development and Industrial Pharmacy, 25 Sep 2018, 44(12):1953-1965 DOI: 10.1080/03639045.2018.1503298)
Thank you for the suggestion, the reference was added to the manuscript.
- On line 71, please use lag time instead of delay time (more common in drug delivery)
We agree with the reviewer. The terminology was changed.
- The sentence on Line 108 should start from the beginning of the line.
The mistake was corrected.
- Line 136, The dry ….the T is missing
The misprinting was corrected.
- On line 165, please use the term free radicals instead of radicals.
We do not consider this rectification necessary because the term radicals was introduced and explained above.
- The materials and methods section still reports results in it, which are supposed to be under results and discussion. Please rectify.
We moved the calibration curve for DPI into Results and discussion. As for the methods of processing the EPR spectra, these methods were described in the literature earlier so it is not the subject for discussion in the present article.
- Justification and discussion about why the researchers chose the film's geometry and not a particle or cylindrical or tablet form of the polymer. Would the EPR have the ability to measure drugs and radicals movement the same in this case with the same accuracy and sensitivity?
The best way to determine the fundamental regularities of the release kinetics determined by the nature of the polymer and the drug is to investigate the structures with easily reproduced geometry – cylindrical forms, tablets or films. The last is the most promising since in this case the diffusion problem can be regarded as a one-dimensional. That is why we studied the release from polymer films. The corresponding explanation was added into Introduction.
Reviewer 3 Report
The author has addressed most of the given suggestions well. I would like to suggest again the following suggestion to incorporate your answer in the introduction section of the revised manuscript with a more elaborative discussion.
"[Highlight the novelty of the current investigation in comparison to your previous work “Paramagnetic bioactive encapsulated in poly(D, L-lactide) microparticles: Spatial distribution and in vitro release kinetics” published recently in Journal of Supercritical Fluids 158 (2020) 104748.]"
The inclusion of this discussion in the introduction section will be helpful for the reader to understand in a better way the similar line of research with the significant outcomes. As the author has indicated in his reply, different perspectives of drug release from PLGA microparticles versus PLGA film. Indeed, the reply needs a more descriptive explanation of the finding and it should be included in the revised manuscript.
Author Response
We agree with the reviewer. The discussion was added into Introduction.